# A Preliminary Analysis of Anthropogenic and Natural Impacts on a Volcanic Lake Ecosystem in Southern Italy by UAV-Based Monitoring

**DOI:** 10.3390/ijerph20010005

**Published:** 2022-12-20

**Authors:** Domenica Mirauda, Maria Giuseppina Padula, Enza Mirauda, Claudio Paternò, Fiorenzo D’Onofrio, Domenico Loguercio

**Affiliations:** 1School of Engineering, Basilicata University, Viale Ateneo Lucano 10, 85100 Potenza, Italy; 2Direzione Generale per le Politiche Agricole, Alimentari e Forestali, Regione Basilicata, Via Vincenzo Verrastro 10, 85100 Potenza, Italy; 3Consultant of Environmental Engineering Services Ltd., Via Varco d’Izzo 28, 85100 Potenza, Italy; 4Consultant of Consorzio di Bonifica della Basilicata, Via Annunziatella 64, 75100 Matera, Italy

**Keywords:** volcanic lake, UAV monitoring, highly detailed spatial data, flooded shores, lake-water level, anthropogenic stressors

## Abstract

Lakes play an important role in providing various ecosystem services. However, stressors such as climate change, land use, or land-cover change threaten the ecological functions of lakes. National and international legislations address these threats and establish consistent, long-term monitoring schemes. Remote sensing techniques based on the use of Unmanned Aerial Vehicles (UAV) have recently been demonstrated to provide accurate and low-cost spatio-temporal views for the assessment of the ecological status of aquatic ecosystems and the identification of areas at risk of contamination. Few studies have been carried out so far on the employment of these tools in the monitoring of lakes. Therefore, high-resolution UAV surveys were used to analyse and evaluate natural and anthropogenic impacts on the habitat status of a volcanic lake in a protected area. Five UAV flights took place during a year-long cycle (November 2020 to November 2021) in a volcanic lake located in southern Italy. For each flight performance, an orthomosaic of georeferenced RGB images was obtained, and the different features of interest were monitored and quantified using automated processing in a GIS environment. The UAV images made it possible not only to estimate the flooded shores but also to detect the impact of human-made structures and infrastructures on the lagoon environment. It has been possible to observe how the rapid changes in lake-water level have led to the submersion of about 90.000 m^2^ of terrain in winter, causing the fragmentation and degradation of habitats, while the connectivity of the natural ecosystem has been threatened by the presence of the road around the lake. The proposed methodology is rather simple and easily replicable by decision makers and local administrators and can be useful for choosing the best restoration interventions.

## 1. Introduction

Lakes play a crucial role in the Earth’s biosphere, although they represent only a small part of the water on our planet [1]. They provide habitats for a wide range of species; form essential components in hydrological, nutrient, and carbon cycles [2]; and offer various ecosystem services to human beings [3]. Among freshwater ecosystems, lakes are considered composite and dynamic structures, each located in a unique landscape context [4]. However, anthropogenic exploitation and multiple interacting stressors threaten ecological functions of inland waters over the entire globe [5]. These stressors can lead to eutrophication, deterioration of water quality, morphological alterations, acidification, or increased water temperatures [6,7]. In addition, due to greater water withdrawal and intensified climate change, many inland lakes around the world are shrinking and will eventually turn into substantial sources of air pollution [8].

Several national and international directives (US Clean Water Act, South African National Water Act, National Water Management Strategy of Australia and New Zealand, the Canada Water Act, and the Water Framework Directive in Europe) aim to improve the ecological state of inland waters by identifying stressors and by implementing sustainable management strategies supported by more or less frequent monitoring, for example [9,10,11]. Currently, most monitoring programmes are field-based, even if sampling and analysis are labour, cost, and time consuming [12]. Despite providing information on a species level, single measurements or unevenly distributed sampling points are problematic and hardly capture the temporal and spatial variability of phenomena such as short-living cyanobacterial or phytoplankton blooms [3,13]. For a comprehensive understanding of lake ecology and the role of lakes “as sentinels, integrators and regulators of climate change” [14], frequent and consistent long-term monitoring approaches are required globally [15,16].

In the last decades, the development of many high-accuracy remote sensing (RS) techniques has allowed for the acquisition of much more reliable data, with a much higher survey frequency [17]. This can be fundamental to the study of water bodies because it enables the researcher to perform multi-scale analyses that are crucial to the understanding of the complex connection among the dynamic factors influencing the water status. Among the different RS techniques, the one based on the UAV employment has seen an increasing interest in the rapid and low-cost capturing of ultra-high spatial resolution monitoring data in water environments [18]. It is also considered a rather simple and easily replicable tool for non-experts in specific surveying technologies and processing of derived data. The clear advantages of this approach are cost-effectiveness and flexibility. Furthermore, UAV surveys can now be considered a necessary intermediary between in situ measurements and satellite remote sensing because they allow rapid deployments and easy control of flight paths and altitudes to capture different scales of maps within a short time. However, UAVs are still relatively new in the research field of lake monitoring.

Therefore, this study could potentially be one of the few examples of a UAV mapping approach to investigating the influence of climate change and human activities on a lagoon environment. In particular, the aim of this paper is to test if low-cost tools provide suitable and rapid methods for identifying and quantifying natural and anthropogenic impacts on the spatio-temporal evolution of the vegetation cover around and within the lake.

The investigated lake is located in north-west Basilicata, southern Italy, and it is one of the most important Italian lakes of volcanic origin. In the last few years, random field samplings have highlighted the disappearance of some significant habitats caused by inappropriate area management and by the effects of climate change. Under these conditions, there is an evident need to devise and implement a remote sensing system for the regular monitoring of the lagoon environment and to respond rapidly to its changes, based on such a flexible and operative tool as UAV surveys. Taking into account the very complex set of environmental issues and their intricate spatio-temporal patterns, the present study examines the possibilities, specific features, and applications of UAV-based environmental monitoring of the area around Lago Grande. It would represent an initial and important elaboration stage of a multi-scale and multi-temporal remote sensing scheme for analysing a highly dynamic and vulnerable area.

Examinations of multiple aerial surveys on Lago Grande between November 2020 and November 2021 revealed rapid changes in lake-water level and quantified the anthropogenic impact. This result has highlighted how the use of low-cost survey techniques could help to rapidly detect the stressors acting on the indigenous fauna and flora in order to choose the interventions and measures able to hinder such factors and protect biodiversity.

## 2. Materials and Methods

### 2.1. Study Area

Lago Grande is located in the northwest of Basilicata, within the caldera of Monte Vulture between the municipalities of Rionero in Vulture and Atella. It is of volcanic origin and is within the perimeter of the Special Conservation Area (SAC) of “Monte Vulture”—IT9210210, a site of Rete Natura 2000 managed by the Regional Natural Park Authority of Vulture since 2017.

Lago Grande represents one of the most prominent European terrestrial paleo environmental archives, with a continuous 100 kyr record spanning the Eemian interglacial up to the modern industrial period [19]. Although it is a site rich in habitats of community interest and a popular destination for tourists from all over the world, at present it is still little investigated. In recent decades, it has been subjected to various natural and human stresses, which have been threatening the important flora and fauna species. For this reason, monitoring and studying the lake and the surrounding environment has become essential.

Lago Grande is in the Ofanto basin, which covers an area of 2780 km^2^. The river Ofanto originates from Campania, crosses Basilicata and Puglia, and flows into the Adriatic Sea. In Figure 1, the Lucanian Ofanto basin, where Lago Grande is located, is shown. 

Lago Grande covers 0.41 km^2^ and is at about 650 m above sea level. It has an elliptical shape and a funnel-shaped cavity, with a maximum depth of about 40.0 m and a volume of 3402,183.2 m^3^. Fed by groundwater and by Lago Piccolo through an artificial channel, Lago Grande flows into the river Ofanto through the Torrente Laghi channel.

Lago Grande can be classified as meromictic. Its deep waters, starting from 25 m, remain anoxic and are never able to mix completely with the superficial waters due to the permanent chemical stratification [20]. The waters of the layer from 0 to 25 m, on the other hand, stratify during the spring, remain stratified in the summer and autumn, and then are completely mixed in the winter with uniform temperatures throughout the water column.

An analysis of Corine Land Cover data from 1990 and 2018 in the Lucanian Ofanto basin shows a change in land use recorded in the last thirty years (Table 1). As detailed in the table, where only the land use typologies with percentages over 0.5% are reported, residential and commercial areas have grown slightly over the years. Furthermore, arable land on non-irrigated land and temporary crops associated with permanent crops have given way to complex crop and parcel systems, and to areas mainly occupied by agricultural crops. There has also been an increase in vineyards, olive groves, and chestnut groves. In addition to these, mainly conifer plantations and riparian poplar forests, as well as turkey oak, frainetto, and oak forests, are recorded (Table 2). The deciduous forests that were planted on reclaimed areas in the 1990s are now partly replaced by evolving woodland and shrub vegetation.

The area is rich in tree species such as the *Alnus glutinosa*, the *Salix alba*, the *Salix caprea* L., the *Fraxinus excelsior* L., the *Populus alba* L., the *Populus nigra* L., the *Populus tremula* L., the *Cucubalus baccifer* L., the *Cyperus fuscus* L., the *Eupatorium cannabinum* L., the *Humulus lupulus* L., the *Juncus effusus*, the *Lathraea squamaria* L., the *Ligustrum vulgare*, the *Rubus caesius* L., and the *Solanum dulcamara* L.

In the wetter stagnations, alien species such as ash trees are observed, while *alders* and *willows* are found near the water. On the edge of the lake, there are humid grasslands, albeit with very limited coverage, consisting mainly of Mediterranean rushes.

The lake also shows aquatic and hygrophytic vegetation of great naturalistic value, which is able to favour the spontaneous purification processes of the water. This vegetation includes the *Phragmites australis*, the *Typha latifolia*, the *Nymphaea alba*, the *Potamogeton natans*, the *Potamogeton pectinatus*, the *P. lucens*, the *Myriophyllum spicatum*, the *Nitellopsis obtuse*, and the *Ceratophyllum demersum*. The waters of the lake have particularly valuable fish species such as the *Dianthus vulturius* and the *Alburnus vulturius*, which in recent decades have been threatened by the presence of the Perch Trout.

In the natural and artificial open channels, the otter (*Lutra lutra* species) is found, a mammal that has now become extremely rare and is an endangered species. In the reeds, there are numerous species of birds such as the moorhens (*Gallinula chloropus*), the water rails (*Rallus acquaticus*), the reedbats (*Acrocephalus arundinaceus*), the reedbugs (*Acrocephalus scirpaceus*), the long-tailed tits (*Aegithalos caudatus*), the nightingales (*Luscinia megarhynchos*), the whiskers (*Panurus biarmicus*), the cormorants (*Phalacrocorax carbo*), the mallards (*Anas platyrhynchos*) and teals (*Anas crecca*). Coots (*Fulica atra*), grebes (Podiceps cristatus), and little grebes (*Tachybaptus ruficollis*) migrate in winter from Lago Grande to Lago Piccolo. The Eagle Owl (*B. bubo* species), the Black Kite (*M. migrans* species), the Honey Buzzard (*P. apivorus* species), the Marsh Harrier (*C. aeruginosus* species), and the Red Kite (*M. milvus* species) nest on the rocks.

Finally, a unique species lives in the area: the *Brahmaea europaea*, a butterfly with a wingspan of over seventy millimetres. It flies on only a few days of the year and for a few hours during the day, even with snow. It is considered a living fossil and a Miocene relic [21].

### 2.2. UAV Images Acquisition

As shown in Figure 2, topographic surveys were performed with a commercial DJI Phantom 4 Professional quadcopter, which was used to monitor the lake and surrounding environment. The gross weight of the aircraft is 1380 g, and it has an endurance of 30 min when flying at a maximum speed of 57 km/h. The UAV is equipped with an autopilot, which is constituted of an artificial heading and attitude reference system (AHRS) based on the combination of 3-axis accelerometers and rate gyros, a magnetometer, static/dynamic pressure, and a GPS. This system allows fully autonomous navigation, following a user-defined flight plan. It is equipped with a 1-inch 20-megapixel sensor capable of capturing 4k/60 fps video and burst-mode stills at 14 fps and is assisted by a high-accuracy differential GNSS equipped with Real Time Kinematic (RTK) technology for corrections. This model of aircraft was used for this research because it has the same great impact on the users that the previous Phantom DJI multi-copters had, and it is also the most versatile and easy-to-use lightweight RTK DJI multi-copter. The addition of a GNSS antenna–receiver module to enable the DJI-P4RTK to kinematic GNSS [22] with centimetre-level accuracy has, thus, made this aircraft of great interest in the field of lagoon mapping with direct georeferencing.

The images acquired with the drone have been oriented, scaled, and georeferenced thanks to the placement on the ground of a series of fixed and mobile markers, known as Ground Control Points (Figure 3), which were homogeneously distributed on the surface to be monitored at different altitudes. These were measured with a GNSS Topcon HiPer Pro of millimetric precision and a total station located in the municipality of Poggiorsini (Bari).

The flight plan was carried out to obtain a good coverage of the investigated area and a high quality of images by selecting the distance from the UAV to the ground surface (Ground Sampling Distance). The images were collected by using the single grid option of the autopilot software with a GSD equal to about 90 m (corresponding to 3.8 cm/pixel resolution) and with a 75% overlap between the footprints of the images. A constant velocity of 4 m/s was chosen to reduce the effects of solar radiation and the movement of the leaves due to the wind. All flights were planned with a freeware application and performed in automatic mode in order to guarantee the high quality of the survey. For each survey, the Root Mean Square Error (RMSE) of elevation data was calculated (Table 3).

The licensed software Agisoft Metashape Professional 1.5.1 [23] was used to reconstruct the Digital Surface Models (DSM) of the surveyed environments, thanks to the Structure from Motion (SfM) algorithm [24]; this was followed by the definition of very highly accurate orthophotographs, composed of a mosaic of single orthogonal frames (orthomosaics). Both DSMs and orthophotographs/mosaics were shaped with a resolution of 0.05 m and referenced to a global coordinate UTM system. Figure 4 shows the orthomosaics of all UAV surveys.

## 3. Results

### 3.1. Natural Stressors

Lago Grande is characterised by highly heterogeneous ecosystems threatened constantly by various natural stressors. Among the natural pressures, the seasonal fluctuation of the lake level is the most significant one, as it causes erosion of the shore and burial of the lake, as well as flooding phenomena that lead to the destruction of trees and shrubs.

Figure 5 shows an estimate of the submerged areas that was obtained via diachronic comparison between the lakeshores through orthophotos acquired from November 2020 to November 2021. If the lake surface measured in the summer season (6 July 2021) is considered as the baseline, it is possible to quantify that the flooded areas at the beginning of 2021 were around 90,000 m^2^ and about 31,000 m^2^ in spring months. The increase in the lake-water level is due to short and intense rainfall which occurred between November 2020 and May 2021. Figure 6 shows the monthly total rainfall recorded in a station near the lake during the monitoring period. Not only has the rapidly raised water level led to serious issues of shore submergence and flooding of the coastal zone, but it has also caused a fragmentation and degradation of riparian vegetation, as observed in Figure 7a–c.

In addition, Figure 7d also shows the effect of the flooding, which lead to a separation of a large piece of shore with all its vegetation that shifted from one bank to the other due to wind action. Moreover, the accumulation of sediments along the Torrente Laghi, caused by the intense and short rainfalls, could also contribute to the hindering of the passage of fish fauna from the Ofanto River to the lake and vice versa (Figure 7e). The surface runoff from arable land during rainfall raises the concentration of phosphorus (>60 mg/L) and ammonia nitrogen (>10 mg/L), as recorded in previous years [20]. This could worsen the lake-water quality and cause the growth of both algal populations and floating plants, especially in summer months.

Figure 7f, which was acquired by UAV, shows the formation of the surface hydrographic network in arable lands surrounding the lake.

In recent years, the increase in air temperatures, as recorded in a station near the Lago Grande (Figure 8), has kept the surface waters of the lake warm even during the autumn season, thus favouring the permanence of large algae and cyanobacteria with reduced palatability to zooplankton. Figure 9 shows the presence of green algae, which may indicate a potential eutrophication process. This condition can affect the structure of planktonic community with cascading effects through the whole food web and even on ecosystem processes.

### 3.2. Anthropic Stressors

The negative effects of anthropogenic activities began in the early 1900s with the construction of reclamation works and of settlements for civil and tourist uses in the 1950s, with roadways asphalted and cemented later. All of this construction has contributed to the fragmentation of ecosystems and the reduction of particularly valuable habitats.

Figure 10 shows the road around the lake, the connecting roads, the buildings, and the impermeable surfaces depicted in the orthophoto from 16 November 2020. As can be seen from the figure, the element that has the greatest impact on the lagoon ecosystem is the road around the lake. This road has a surface area of approximately 21,000.00 m^2^, which is comparatively larger than the connecting roads (1400.00 m^2^), buildings (6940.00 m^2^), and impermeable surfaces (13,030.00 m^2^). The impact of the road is due to a combination of car traffic; the harmful substances it releases into the air, that is, pollutants and dust deposits; increased sound and noise intensity, which alter the levels of natural light in the nocturnal environment; and the frequent crushing of animals on the roadway and on car windshields, leading to the rarefaction or disappearance of plant and animal populations over time.

The UAV photos show the presence of some structures built along the Torrente Laghi channel, which constitute a significant interruption of the ecological connection between the lake and the Ofanto (Figure 11). In particular, along the main stream, there are narrowings and gates which were once used for irrigation purposes and which no longer allow the young eels (*Anguillidae*) to go up towards the lake.

Furthermore, the fish fauna is also seriously threatened by authorised fishing in the lake, which often does not respect the rules of the protected area, and by the presence of uncontrolled tourism, which causes the trampling of the lakeside vegetation and the abandonment of waste.

In addition, the navigation waves cause a resuspension of the bed sediment and make the spontaneous plant repopulation on the banks, which suffer continuous erosion, more difficult. Finally, the contamination by oils, fuels, and antifouling paints can lead to fish death.

## 4. Discussion

The results obtained by a multi-temporal UAV data analysis with ultra-high spatial resolution underlined the presence of flooded sites which, in time, led to the rapid degradation of the lagoon habitat. This intensified the lake eutrophication and provoked hazardous algal blooms, thus creating favourable conditions for phytoplankton development, as demonstrated by recent studies in the literature [18,25]. Furthermore, the increase of the water level induced a non-uniform erosion and sediment accumulation phenomenon along the shores, which caused the fragmentation of important habitats and lead to their disappearance.

The orthophotos have also shown important elements of discontinuity for the lagoon ecosystem caused by the roads around lake, as well as interruptions of the ecological connection between the lake and the Ofanto due to the presence of narrowings and gates along the Torrente Laghi channel, which hinder the return of young eels of the family of *Anguillidae*.

The UAV-based methodology here proposed required much less time and expense than the application of traditional acquisition methods. In fact, in previous years, the same authors measured the lake surface through a topographic survey. This type of approach, which is an integral part of routine environmental monitoring practices already handled by related authorities, not only required a full day of work with costs around 6000.00 euros but also had incomplete results at times, since some areas were inaccessible even from the ground. Therefore, to solve this problem, the land measurements were integrated with those taken from the lake by boat. This has further increased the cost and time of operations by about 30%.

The UAV-based methodology, instead, makes it possible to obtain a high-resolution orthoimage by flight-planning the final product in almost real time and at very affordable costs (around 1500.00 per flight). For example, while field surveys usually take a whole day, information collection via remote sensing requires no longer than 30 min, which also avoids potential issues related to changing weather conditions and, thus, its impact on the quality of the results. In general, UAV methods are also not as invasive as traditional surveys, which helps to preserve the lakesides integrity. In this study, for example, flight operations were performed from the lake shore, and the short duration of drone flights barely impacted the environment. By contrast, traditional surveys often take several hours and cause greater disturbance to the lake ecosystem (such as the trampling of the lakeside vegetation) than UAV flights. Another advantage of the small-sized UAVs used in this study was that they did not seem to disturb the birds either.

Regarding the limitations of the proposed UAV-based method, the following need to be considered: (1) there can be restrictions from national regulations on drone use; (2) weather conditions might affect the geomatic product quality; (3) when assessing the temporal evolution of a water body, intrinsic aspects might change between flights and influence the final result; and (4) there might be other constraints, such as obstacles or barrier elements, waves disturbing the water surface, or administrative limitations in preserved areas related to no-fly zones or to the height and distance of flight, which need to be accounted for in the planning phase. For instance, in Natural Protected Areas (including waterfowl breeding areas, sensitive wetlands for waterfowls, and ZEPAs), drones are not generally allowed, and authorisation for scientific proposes needs to be obtained.

In the present study, the main limitation recorded was wind velocity; since wind tolerance depends mainly on the equipment size and weight, a higher wind velocity than the one experienced here (2 m/s) could produce blurry images. Especially in the autumn and winter months, flights were hindered on 20% of the days due to adverse weather conditions, such as high wind velocity and intense rainfall events, along with loudness and lighting. The optimal time to perform flights was found to be noon, as recommended by Ortega-Terol et al. [26]. Another limitation was water-surface stabilisation due to either the formation of waves by the wind or the displacement of water birds, which sometimes compromised the image quality, but that was generally avoided thanks to short flight times. Finally, the presence of thick vegetation along the lakeshores during the spring and summer months made the measurements more difficult than that of the lake outline, requiring the modification of the flight plan with local inspections in a non-automatic mode and with a GSD of about 2–3 m.

Considering both the advantages and disadvantages of UAV surveys, it can be concluded that the environmental monitoring of complex and highly dynamic areas based on these tools could become a necessary intermediary between field measurement and satellite data. This approach is rather simple and easily replicable by decision makers and local administrators who, when in need of evaluating the stressors acting on the ecosystem, would thus be able to choose the best interventions and measures to hinder such factors and protect the biodiversity.

Further developments of the present research should involve the use of UAVs equipped with multi-spectral cameras to better investigate the impacts of humans and climate change on the native aquatic vegetation of the lake and the surrounding environment, as well as to monitor the health status of the important natural habitats. Another interesting aspect to be dealt with in the future would be the integration of UAV monitoring with the assessment of the lake-water quality, which would help to better understand how the presence of inorganic and organic contaminants can favour hazardous algal blooms, so creating the conditions for the phytoplankton development and the progress of the eutrophication phenomenon. Finally, to better understand the impacts of climate change on the lagoon ecosystem, it would be useful to research a relationship between the lake’s areal evolution by UAV in different years and with varying temperature and precipitation trends, thanks to the development of a database of orthophotos and weather data.

## 5. Conclusions

The results obtained with this study are as follows:

(1)The multi-temporal UAV data analysis revealed rapid changes in the lake-water level and made it possible to estimate the submerged and flooded shores. These areas, which reach sizes of up to about 90,000 m^2^ during winter, are one of the main causes of the fragmentation and degradation of habitats, as reported by different studies in the literature.(2)The UAV images showed the presence of hazardous algal blooms and cyanobacteria in autumn months due to the increase of temperatures, thus creating favourable conditions for phytoplankton development.(3)The orthophoto analysis made it possible to quantify the impact of structures and infrastructures on the environment surrounding the lake. In particular, the road around the lake is a major source of discontinuity in the landscape and the ecological network, preventing or partially limiting the connectivity of the natural ecosystem.(4)The proposed monitoring methodology would be useful for planning measures and interventions aimed at restoring the native riparian and aquatic environments, as well as regulating or prohibiting activities with a high negative impact on the flora and fauna species.

## Figures and Tables

**Figure 1 ijerph-20-00005-f001:**
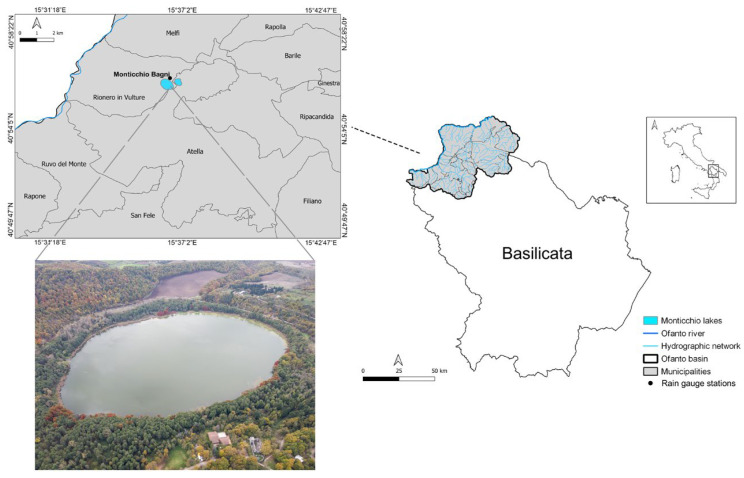
Lucanian Ofanto basin and aerial photo of Lago Grande.

**Figure 2 ijerph-20-00005-f002:**
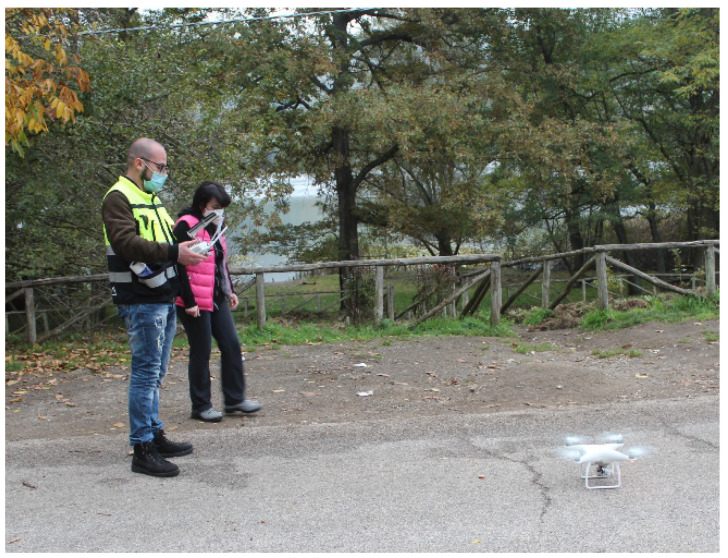
Dji Phantom 4 used for surveys.

**Figure 3 ijerph-20-00005-f003:**
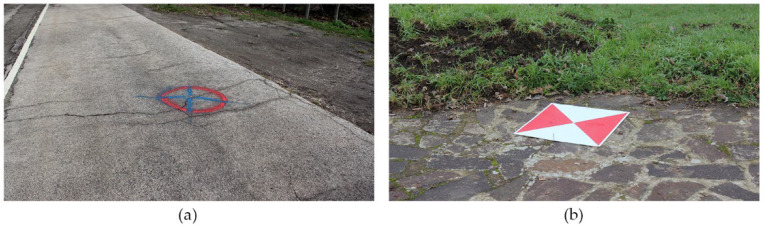
(**a**) Fixed and (**b**) mobile markers.

**Figure 4 ijerph-20-00005-f004:**
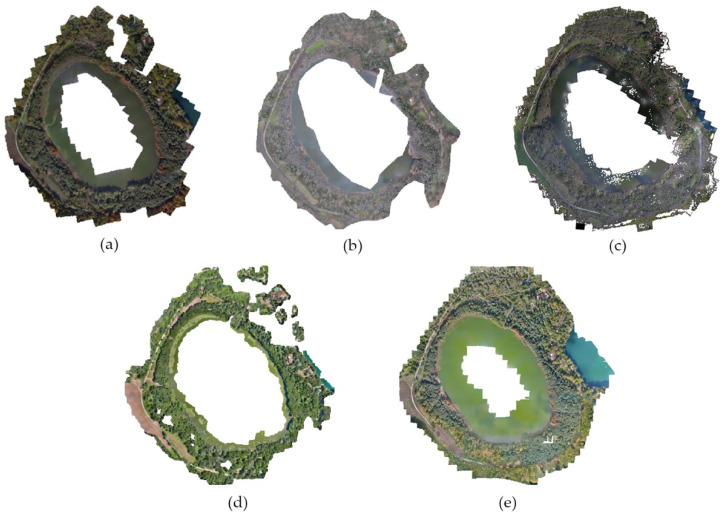
Orthophotos of: (**a**) 16 November 2020; (**b**) 29 January 2021; (**c**) 15 April 2021; (**d**) 6 July 2021; (**e**) 10 November 2021.

**Figure 5 ijerph-20-00005-f005:**
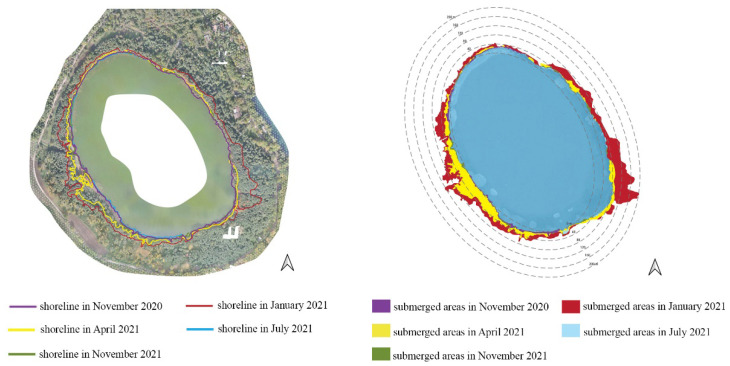
Diachronic comparison between the lakeshores.

**Figure 6 ijerph-20-00005-f006:**
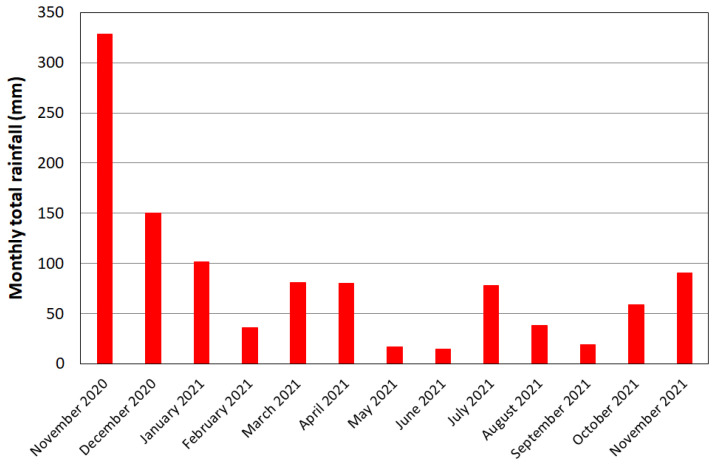
Monthly total rainfall recorded in a station near the lake.

**Figure 7 ijerph-20-00005-f007:**
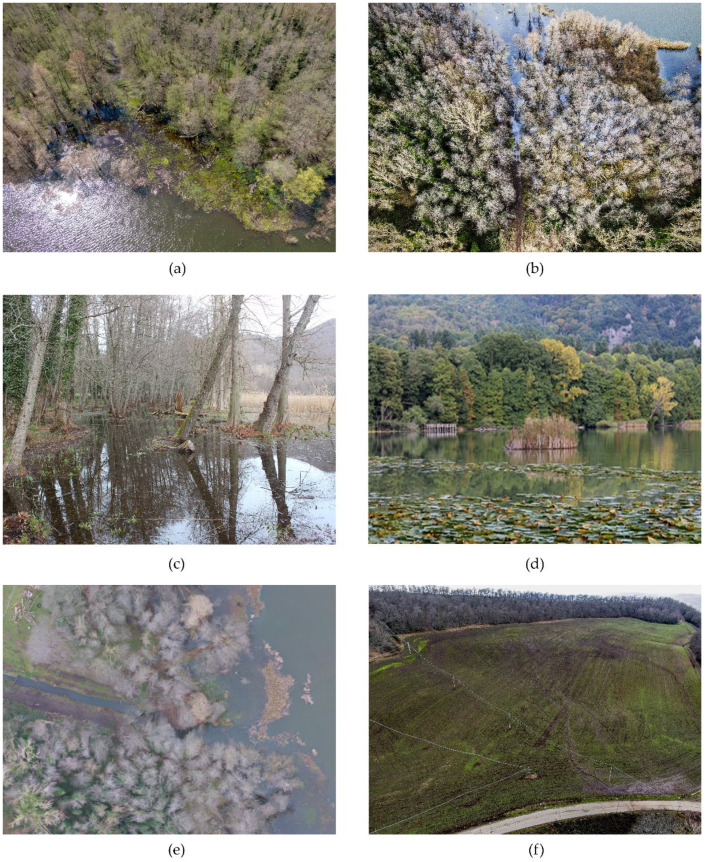
Flooded coastal areas.

**Figure 8 ijerph-20-00005-f008:**
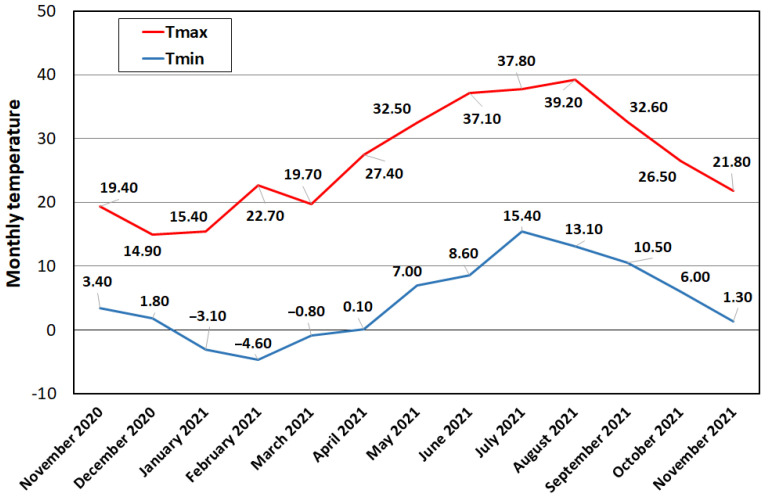
Maximum and minimum temperatures recorded at the Monticchio Bagni Station from November 2020 to November 2021.

**Figure 9 ijerph-20-00005-f009:**
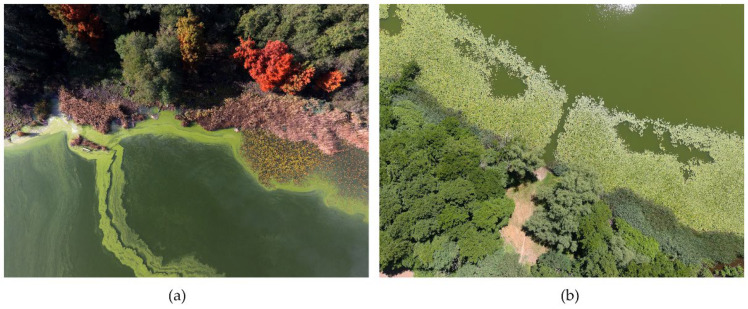
Algal blooming acquired by UAV.

**Figure 10 ijerph-20-00005-f010:**
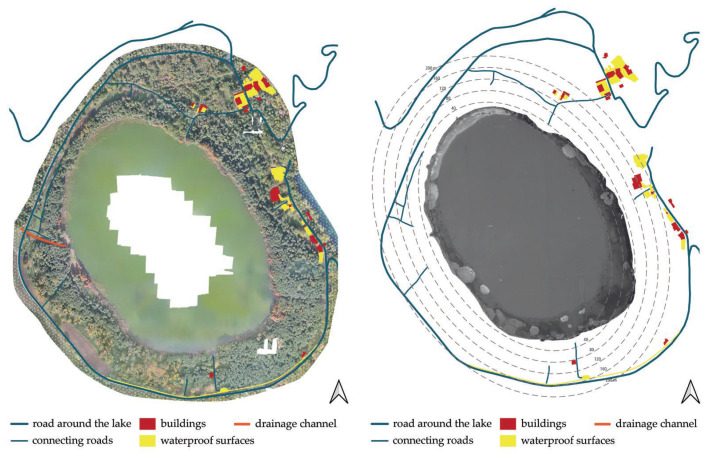
Structures and infrastructures in the area surrounding the lake.

**Figure 11 ijerph-20-00005-f011:**
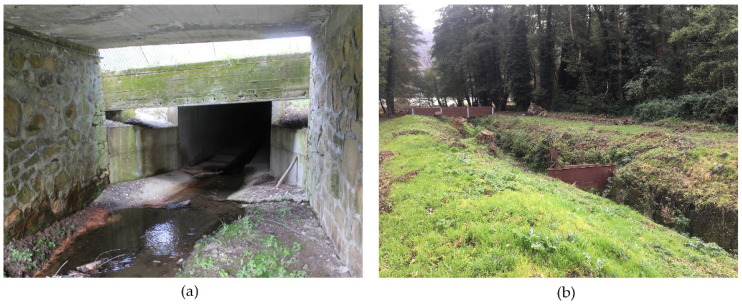
Constructions along the Torrente Laghi channel.

**Table 1 ijerph-20-00005-t001:** Percentages of land use between 1990 and 2018 calculated in the area of the Lucanian Ofanto basin.

Typologies	1990	2018	2018–1990
Continuous urban fabric	0.26%	0.73%	0.47%
Discontinuous urban fabric	1.07%	2.14%	1.07%
Industrial or commercial units	0.92%	2.10%	1.18%
Mineral extraction sites	0.04%	0.21%	0.17%
Non-irrigated arable land	19.37%	16.54%	−2.83%
Vineyards	2.99%	6.21%	3.22%
Fruit trees and berry plantations	0.16%	0.70%	0.54%
Olive groves	9.90%	11.93%	2.03%
Pastures	1.11%	1.91%	0.80%
Annual crops associated with permanent crops	10.23%	0.13%	−10.10%
Complex cultivation patterns	5.00%	11.10%	6.10%
Land principally occupied by agriculture with significant areas of natural vegetation	4.33%	16.14%	11.81%
Broad-leaved forest	36.49%	12.15%	−24.34%
Coniferous forest	0.04%	0.60%	0.56%
Mixed forest	0.80%	1.32%	0.52%
Natural grasslands	3.19%	4.95%	1.76%
Sclerophyllous vegetation	0.70%	0.12%	−0.58%
Transitional woodland-shrub	2.62%	9.60%	6.98%
Beaches—dunes—sands	0.04%	0.20%	0.16%
Inlands marshes	0.24%	0.40%	0.16%
Water bodies	0.50%	0.80%	0.30%

**Table 2 ijerph-20-00005-t002:** Percentages of Corine Biotopes habitats in 2013 calculated in the area of the Lucanian Ofanto basin.

Typologies	2013
Fresh waters	0.15%
Eastern sub-Mediterranean white oak woods	1.47%
Southern Italian quercus frainetto woods	5.06%
Chestnut woods	1.15%
Quarries	0.17%
Southern Italian quercus cerris woods	7.10%
Middle-European rich-soil thickets	1.21%
Towns	1.11%
Extensive cultivation	12.28%
Southern Italian beech forests	0.72%
Mediterranean riparian poplar forests	1.37%
Fruit orchards	0.34%
Mediterranean gravel beds	0.19%
Italian supra-Mediterranean holm-oak forests	0.07%
Oleo-Ientisc brush	0.05%
Olive groves	4.63%
Conifer plantations	0.55%
Eucalyptus plantations	0.05%
Sub-Mediterranean Mesobromion	1.98%
Central and southern Apennine dry grasslands	2.11%
Middle-European Brachypodium-dominated semi dry grasslands	0.15%
Mesophile pastures	0.80%
Medlterranean subnitrophilous grass communities	4.01%
Southern Italian and Sicilian quercus pubescens woods	0.31%
Unbroken intensive cropland	49.31%
Active industrial sites	0.56%
Reed beds	0.10%
Tyrrhenian sub-Mediterranean deciduous thickets	1.39%
Vineyards	0.89%

**Table 3 ijerph-20-00005-t003:** RMSE of elevation data for each survey resulting from the validation process.

Survey	RMSE (m)
16 November 2020	0.001
29 January 2021	0.025
15 April 2021	0.007
27 July 2021	0.048
10 November 2021	0.023

## Data Availability

Not applicable.

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
