# Peer review of "A Preliminary Analysis of Anthropogenic and Natural Impacts on a Volcanic Lake Ecosystem in Southern Italy by UAV-Based Monitoring"

_ijerph, 2022, doi:10.3390/ijerph20010005_

Round 1

Reviewer 1 Report

I would like to congratulate the authors on their work. I think it is a very relevant topic that will attract more and more attention from researchers.

General comments:

The authors have presented only the advantages of using drones in monitoring lakes. Does this type of methodology have disadvantages? What are they?

Several English expressions are unusual, suggesting the authors may have used an automatic translator. Therefore, I suggest the manuscript should be reviewed by a native English speaker.

The conclusion and the discussion need to be completely rewritten. Authors have difficulty understanding the difference between a discussion and a conclusion. The conclusion of the manuscript is 1.5 times longer than the discussion. I have never seen something similar.

The first paragraph of the conclusion can be used as an abstract since it is a synopsis of the study. However, the conclusion is not a synopsis. The conclusion should address the main research question and propose a straightforward and concise answer. The main question is “is to test if low-cost tools are suitable and rapid methods to identify and quantify the natural and anthropic impacts on spatio-temporal evolution of the vegetation cover around and within the lake” and the conclusion should focus on responding to this question.

Several parts of the conclusion should be in the discussion (ex. from line 360 until line 376).

Specific comments:

Lines 128 e 1290: The author state: “The waters of the layer from 0 to 25 meters, on the other hand, stratify during the spring, remain stratified in the summer and autumn and then are completely mixed in the winter.”

Then they state: “The temperatures are uniform throughout the water column between December and February, while for the rest of the year they mix”

In my understanding, if the water column is uniform, it means the lake is mixed. If the temperature is uniform in two months and the lake mixes in the remaining months, then it never stratifies. The two sentences are contradicting each other.

Lines 141-145: Why the authors are calling the tree species “habitats”?

Line 165: Please specify the otter species or at least the genus.

Lines 173-174: Please, include species names.

Lines 175-178: The authors report several species, including endemic and migratory species, without citing any scientific studies that support these statements. The studies from which this information was taken must be cited.

Line 233: replace the expression “it is possible to observe” with “it was possible to quantify (or measure)”

Line 245: Replace “Besides” with “In addition”

Line 261: Replace “unappetizing to zooplankton” with “with reduced palatability to zooplankton”

Lines 261-262: Eutrophication is a complex process. Although the bloom of green algae may be evidence of this process, it is not the only one. Thus, one cannot conclude about eutrophication, based solely on these photos. I suggest stating “presence of green algae, which may indicate a potential eutrophication process”

Lines 262-246: Suggestion for the last sentence: “This condition can affect the structure of planktonic community with cascading effects through the whole food web and even on ecosystem processes.”

Line 270: I suggest replacing “Anthropic” with “Anthropogenic” throughout the whole text.

Line 276: Replace “waterproof surfaces” with “impermeable surfaces” throughout the whole text.

Line 289: Specify the species or at least the genus of the eels.

Lines 290-293: There is no reason for these two paragraphs to consist entirely of a single sentence. Please restructure the text accordingly.

Lines 313-316: The authors state “Also the increase of air temperatures, recorded in the summer and autumn months of the last years, has kept the surface waters of the lake warmer than usual, influencing the distribution of aquatic organisms and, consequently, the trophic relationships in ecosystem of Lago Grande.”

Although they presented air temperature values, they did not collect any data in this regard (water temperature, plankton community structure, etc). Thus, this statement is completely speculative and should be taken out of the discussion, or the authors should cite studies that support this statement.

Lines 316-324: The authors develop a whole complex explanation to justify a possible result that was hypothesized (they did not measure it). This part can be completely deleted, as it does not concern the discussion of the results.

Figure 3: I believe this map can be removed, as it adds no new information beyond what is already described in the text and in table 1. Perhaps the authors should highlight only the land uses that have changed the most because the visual and qualitative evaluation does not allow us to identify major changes.

Table 1: Please add a third column with the difference between the percentages of land use between the two years.

Figure 4 is very beautiful but unnecessary.

Figure 13: Is not possible to read the legend.

Figure 14: Replace “Works” with “Constructions”

Author Response

We are very grateful for your careful reviews and your useful comments. We benefited a lot not only for the technical points, but also for the presentation of the paper, which has now been improved according to your advice. 

Below, we are now addressing the comments point by point, according to your suggestions. The parts in red are our responses.

General comments:

The authors have presented only the advantages of using drones in monitoring lakes. Does this type of methodology have disadvantages? What are they?

According to the Reviewer’s useful suggestions, we have now described them in the paragraph of discussion.

Several English expressions are unusual, suggesting the authors may have used an automatic translator. Therefore, I suggest the manuscript should be reviewed by a native English speaker.

The language has been completely revised.

The conclusion and the discussion need to be completely rewritten. Authors have difficulty understanding the difference between a discussion and a conclusion. The conclusion of the manuscript is 1.5 times longer than the discussion. I have never seen something similar. The first paragraph of the conclusion can be used as an abstract since it is a synopsis of the study. However, the conclusion is not a synopsis. The conclusion should address the main research question and propose a straightforward and concise answer. The main question is “is to test if low-cost tools are suitable and rapid methods to identify and quantify the natural and anthropic impacts on spatio-temporal evolution of the vegetation cover around and within the lake” and the conclusion should focus on responding to this question. Several parts of the conclusion should be in the discussion (ex. from line 360 until line 376).

According to the Reviewer’s useful suggestions, we have now reorganized the conclusion, moving some sentences to the discussion, and we have focused on key points. We hope to have now satisfied the Reviewer’s request.

Specific comments:

Lines 128 e 1290: The author state: “The waters of the layer from 0 to 25 meters, on the other hand, stratify during the spring, remain stratified in the summer and autumn and then are completely mixed in the winter.”

Then they state: “The temperatures are uniform throughout the water column between December and February, while for the rest of the year they mix”

In my understanding, if the water column is uniform, it means the lake is mixed. If the temperature is uniform in two months and the lake mixes in the remaining months, then it never stratifies. The two sentences are contradicting each other.

We have now modified the sentence.

Lines 141-145: Why the authors are calling the tree species “habitats”?

We have now changed “habitats” to “tree species”.

Line 165: Please specify the otter species or at least the genus.

We have now specified the otter species.

Lines 173-174: Please, include species names.

According to the Reviewer’s suggestions, we have now included species names.

 Lines 175-178: The authors report several species, including endemic and migratory species, without citing any scientific studies that support these statements. The studies from which this information was taken must be cited.

We have now added a reference in the text.

 Line 233: replace the expression “it is possible to observe” with “it was possible to quantify (or measure)”

According to the Reviewer’s suggestions, we have now replaced the expression in the text of the manuscript.

Line 245: Replace “Besides” with “In addition”

According to the Reviewer’s suggestions, we have now made the change.

Line 261: Replace “unappetizing to zooplankton” with “with reduced palatability to zooplankton”

According to the Reviewer’s suggestions, we have now made the substitution.

Lines 261-262: Eutrophication is a complex process. Although the bloom of green algae may be evidence of this process, it is not the only one. Thus, one cannot conclude about eutrophication, based solely on these photos. I suggest stating “presence of green algae, which may indicate a potential eutrophication process”

According to the Reviewer’s suggestions, we have now modified the sentence.

Lines 262-246: Suggestion for the last sentence: “This condition can affect the structure of planktonic community with cascading effects through the whole food web and even on ecosystem processes.”

According to the Reviewer’s suggestions, we have now changed the last sentence.

Line 270: I suggest replacing “Anthropic” with “Anthropogenic” throughout the whole text.

According to the Reviewer’s suggestions, we have now replaced “Anthropic” with “Anthropogenic” throughout the whole text.

Line 276: Replace “waterproof surfaces” with “impermeable surfaces” throughout the whole text.

According to the Reviewer’s suggestions, we have now replaced “waterproof surfaces” with “impermeable surfaces” throughout the whole text.

Line 289: Specify the species or at least the genus of the eels.

We have now specified the genus. 

Lines 290-293: There is no reason for these two paragraphs to consist entirely of a single sentence. Please restructure the text accordingly.

We have now joined the lines as follows:

"Furthermore, the fish fauna is also strongly threatened by authorised fishing in the lake, which often do not respect the rules of the protected area, as well as by the presence of uncontrolled tourism, which causes the trampling of the lakeside vegetation and the abandonment of waste."

Lines 313-316: The authors state “Also the increase of air temperatures, recorded in the summer and autumn months of the last years, has kept the surface waters of the lake warmer than usual, influencing the distribution of aquatic organisms and, consequently, the trophic relationships in ecosystem of Lago Grande.”

Although they presented air temperature values, they did not collect any data in this regard (water temperature, plankton community structure, etc). Thus, this statement is completely speculative and should be taken out of the discussion, or the authors should cite studies that support this statement.

We have now eliminated these sentences in the discussion section and hope to have satisfied the reviewer’s request.

Lines 316-324: The authors develop a whole complex explanation to justify a possible result that was hypothesized (they did not measure it). This part can be completely deleted, as it does not concern the discussion of the results.

According to the Reviewer’s suggestions, we have now deleted this part.

Figure 3: I believe this map can be removed, as it adds no new information beyond what is already described in the text and in table 1. Perhaps the authors should highlight only the land uses that have changed the most because the visual and qualitative evaluation does not allow us to identify major changes.

We have now removed said map.

Table 1: Please add a third column with the difference between the percentages of land use between the two years.

According to the Reviewer’s suggestions, in Table 1 we have now added a third column with the difference between the percentages of land use between the two years.

Figure 4 is very beautiful but unnecessary.

According to the Reviewer’s suggestions, we have now removed Figure 4.

Figure 13: Is not possible to read the legend.

We have now increased the font size of figure legend.

Figure 14: Replace “Works” with “Constructions”

We have replaced “Works” with “Constructions”.

Reviewer 2 Report

This paper intends to describe an environmental monitoring exercise/approach with the aid of UAV. The application and its findings are promising and useful. The georeferencing is particularly useful as it would assist further investigations and research, particularly in a transdisciplinary manner. The inclusion of the temperature and precipitation data needs to be checked with the dominant trend, and as such efforts should be made to facilitate the provision of a detailed database, based on the obtainable data through the proposed technique. A mention of the feasibility of such an approach, by the authors, may lead to more interest from other researchers.

It is also important to note that some of the features addressed in the proposed technique are integral parts of routine environmental monitoring practices already cared for by related authorities. It would help to make a comparison, perhaps in tabular form, to indicate the similarities and comparisons in terms of efforts, costs, and reliability.

For the future use of other researchers, it would help to dedicate a section to the actual technicalities involved in the UAV procedure and possible problems and shortcomings.

The presented findings of this work are too general in places and the conclusion needs to be more focused on the specific results of this work. 

Author Response

We are very grateful for your careful reviews and your useful comments. We benefited a lot not only for the technical points, but also for the presentation of the paper, which has now been improved according to your advice. 

Below, we are now addressing the comments point by point, according to your suggestions. The parts in red are our responses.

This paper intends to describe an environmental monitoring exercise/approach with the aid of UAV. The application and its findings are promising and useful. The georeferencing is particularly useful as it would assist further investigations and research, particularly in a transdisciplinary manner. The inclusion of the temperature and precipitation data needs to be checked with the dominant trend, and as such efforts should be made to facilitate the provision of a detailed database, based on the obtainable data through the proposed technique. A mention of the feasibility of such an approach, by the authors, may lead to more interest from other researchers.

According to the Reviewer’s useful suggestions, we have now added the feasibility of this approach in the discussion section.

It is also important to note that some of the features addressed in the proposed technique are integral parts of routine environmental monitoring practices already cared for by related authorities. It would help to make a comparison, perhaps in tabular form, to indicate the similarities and comparisons in terms of efforts, costs, and reliability.

According to the Reviewer’s useful suggestions, we have now better specified this aspect in the discussion section.

For the future use of other researchers, it would help to dedicate a section to the actual technicalities involved in the UAV procedure and possible problems and shortcomings.

According to the Reviewer’s useful suggestions, we have now described this in the discussion section.

The presented findings of this work are too general in places and the conclusion needs to be more focused on the specific results of this work. 

According to the Reviewer’s useful suggestions, the conclusion is now more focused on the specific results of this work.

Reviewer 3 Report

Please see the Comments in the PDF file.

Author Response

We are very grateful for your careful reviews and your useful comments. We benefited a lot not only for the technical points, but also for the presentation of the paper, which has now been improved according to your advice. 

Below, we are now addressing the comments point by point, according to your suggestions. The parts in red are our responses.

Lakes play a crucial role in the Earth’s ecosystem, which provides habitat for a wide range of species, form essential components in hydrological, carbon cycles, and offer various ecosystem services to human beings. In this paper, the authors described many changes in the land use cover of the Lago Grande lake. The UAV used to monitor the change of lake-water area and relative many work have been done. However, some issues need to be addressed.

The paragraphs are too many and fragmented in the paper, which needs to be improved.

According to the Reviewer’s useful suggestions, we have now modified some paragraphs and hope the overall layout of the paper will be less fragmented.

Some context about Corine Land Change should have been removed in “Result” section. For example, the lines 130-140 presented the land changes. In addition, this paper will be significantly improved if the transfer processes of various land types are calculated.

We would like to underline that the part on Corine Land Change has always been only in the subsection “Study area”. 

According to the Reviewer’s suggestions, in Table 1 we have now added a third column with the difference in the percentages of land use between 1990 and 2018, which confirms what is written in the text from line 130 to line 140 of the original manuscript.

The author used UAV to monitor many aquatic plants, but they were only pictured. I think this paper will be better if the area and change of these aquatic vegetation or one plant were monitored and calculated using UAV, and trying to discuss its impact on the lake ecology, e.g., Algal blooming. In addition, the author indicated that there was a negative effect of anthropic activities to local lake ecosystem in “3.2. Anthropic stressors”, but there was no any quantitative research and analysis except qualitative descript.

We understand the Reviewer’s request but would like to highlight that the change of this aquatic vegetation or of one plant is difficult to quantify since the monitoring activity lasted only one year. However, this analysis could be carried out in the future by extending the observation period.

In addition, the abstract and discussion sections need to be significantly improved in this paper.

According to the Reviewer’s suggestions, we have now modified the abstract and discussion sections and hope to have satisfied the Reviewer’s request.

2 Specific comments

Lines 109-110: Please add a reference.

We would like to underline that there is no specific paper or study on this matter. The information we included was collected by communicating with the local people and consulting newspaper articles.

Line 112: Delete the “in order to reduce disturbing factors”.

According to the Reviewer’s suggestions, we have now deleted this expression.

Lines 124-126: “Its deep waters, starting from 25 metres, 124 remain anoxic and are never able to mix completely with the superficial ones due to the 125 permanent chemical stratification.” Please add the reference.

We have now added the reference.

The author presented “This condition determines a change in the type and mix of lake planktonic species: a variation whose effects will be transferred over time to all the other connected ecosystem components”, which needed to explain further.

We have modified this sentence, hoping to have made it clearer now.

Line 290: “Furthermore, the fish fauna is also strongly threatened by authorised fishing in the lake, which often do not respect the rules of the protected area.” Whether there is a reference?

We would like to underline that there is no specific paper or study on this matter. The information we included was collected by communicating with the local people and consulting newspaper articles.

Line 292: Another form of disturbance is the presence of uncontrolled tourism, causing the trampling of the lakeside vegetation and the abandonment of waste. Whether there is a reference?

We would like to underline that there is no specific paper or study on this matter. The information we included was collected by communicating with the local people and consulting newspaper articles.

Figure and table:

  1. The Figures 1 and 2 can be merged into one figure.

We have now merged Figure 1 and 2 into one figure.

  1. The unit of RMSE was lacked in table 3.

We have now added the unit of RMSE.

Round 2

Reviewer 3 Report

The manuscript is significantly improved, and I have no any comment.